# Developing scandium and yttrium coordination chemistry to advance theranostic radiopharmaceuticals

Korey P. Carter[1,5], Gauthier J.-P. Deblonde [1,2,5], Trevor D. Lohrey[1,3], Tyler A. Bailey[1,4], Dahlia D. An [1], Katherine M. Shield[1,4], Wayne W. Lukens Jr.[1] & Rebecca J. Abergel[1,4✉]

The octadentate siderophore analog 3,4,3-LI(1,2-HOPO), denoted 343-HOPO hereafter, is known to have high affinity for both trivalent and tetravalent lanthanide and actinide cations. Here we extend its coordination chemistry to the rare-earth cations $Sc^{3+}$ and $Y^{3+}$ and characterize fundamental metal–chelator binding interactions in solution via UV-Vis spectrophotometry, nuclear magnetic resonance spectroscopy, and spectrofluorimetric metal-competition titrations, as well as in the solid-state via single crystal X-ray diffraction. $Sc^{3+}$ and $Y^{3+}$ binding with 343-HOPO is found to be robust, with both high thermodynamic stability and fast room temperature radiolabeling, indicating that 343-HOPO is likely a promising chelator for in vivo applications with both metals. As a proof of concept, we prepared a $^{86}Y$-343-HOPO complex for in vivo PET imaging, and the results presented herein highlight the potential of 343-HOPO chelated trivalent metal cations for therapeutic and theranostic applications.

[1] Chemical Sciences Division, Lawrence Berkeley National Laboratory, Berkeley, CA 94720, United States. [2] Glenn T. Seaborg Institute, Physical and Life Sciences, Lawrence Livermore National Laboratory, Livermore, CA 94550, United States. [3] Department of Chemistry, University of California, Berkeley, CA 94720, United States. [4] Department of Nuclear Engineering, University of California, Berkeley, CA 94709, United States. [5] These authors contributed equally: Korey P. Carter, Gauthier J.-P. Deblonde. ✉email: abergel@berkeley.edu

Nuclear medicine is a rapidly growing field based upon the clinical use of radionuclides for diagnostic and therapeutic purposes[1]. Positron emission tomography (PET) and single-photon emission computed tomography (SPECT) are two of the most common imaging modalities for diagnostic purposes, while targeted-radionuclide therapeutic efforts focus on inducing irreversible DNA damage via the emission of either α-particles, β− particles, or low-energy (Auger) electrons[2]. The full potential of nuclear medicine may be realized with theranostics, wherein a molecular targeting vector is labeled with both a diagnostic and a therapeutic radionuclide that are utilized for concomitant imaging and treatment[3–6]. Ideally, the employed radionuclides are a matched pair, where both are radioisotopes of the same chemical element; however, very few elements have isotope pairs with suitable nuclear decay properties. As a result, current efforts focus on perceived chemical homologs such as $^{68}$Ga as the PET imaging agent mimic of $^{177}$Lu or $^{90}$Y for β− therapy. However, this approach is hindered by the different ionic radii of these elements (CN = 6, 0.62 Å for $Ga^{3+}$; CN = 8, 0.977 Å for $Lu^{3+}$; CN = 8, 1.019 Å for $Y^{3+}$)[7,8], which leads to mismatches in coordination chemistry and varying in vivo behavior[9–11]. The identification of chemically identical (i.e., isotopic) pairs of radionuclides with complementary nuclear properties is valuable for the development of new theranostic agents, and $^{44}$Sc/$^{47}$Sc and $^{86}$Y/$^{90}$Y are two pairs with great promise in this arena[1,12–14].

Both $^{44}$Sc and $^{86}$Y offer several advantages compared to the commonly utilized $^{68}$Ga as diagnostic pairs for $^{177}$Lu or $^{90}$Y, including longer half-lives (3.97 and 14.74 h, respectively), higher resolution PET images (Supplementary Table 2), and preferences for higher coordination numbers[15,16]. The latter is of particular importance as the current gold standard for metal ion chelation in nuclear medicine is the octadentate ligand 1,4,7,10-tetraazocyclododecane-1,4,7,10-tetraacetic acid (DOTA). Despite its frequent use as a radionuclide chelator for preclinical and clinical applications, DOTA has a number of limitations, including poor binding kinetics, which generally necessitate heating for radiochemical complexation[17–19]. An ideal chelator for theranostic applications would feature fast radiolabeling kinetics at room temperature, low toxicity, and in vivo stability (i.e., kinetic and thermodynamic inertness), and ligands featuring hydroxypyridinone moieties are known to meet these

criteria[20,21]. The octadentate siderophore analog, 3,4,3-LI(1,2-HOPO), denoted 343-HOPO hereafter, is known to satisfy all three criteria described above, and is an especially effective theranostic chelator due to its propensity to bind with both trivalent and tetravalent metal cations[22,23]. In addition, density functional theory calculations have highlighted structural deformities for 343-HOPO complexes with endogenous metals, which minimizes in vivo competition for the rare-earth cations ($Sc^{3+}$, $Y^{3+}$) included in 343-HOPO chelates herein[24].

Here we begin our investigations by looking at fundamental $Sc^{3+}$-343-HOPO and $Y^{3+}$-343-HOPO binding interactions in solution via UV-Vis spectrophotometry, nuclear magnetic resonance (NMR) spectroscopy, and spectrofluorimetric metal-competition titrations, and in the solid-state via single crystal X-ray diffraction. Subsequently, we conduct proof-of-concept in vivo PET imaging with a $^{86}$Y-343-HOPO complex. $^{86}$Y was selected over $^{44}$Sc for PET studies as $^{86}Y^{3+}$ is a matched pair with $^{90}$Y and a better chemical match than either $Ga^{3+}$ or $Sc^{3+}$ as a diagnostic partner to $^{177}$Lu, both of which ($^{90}$Y and $^{177}$Lu) are β− emitting isotopes included in treatments approved by the U.S. Food and Drug Administration[25,26]. Moreover, while $Sc^{3+}$ and $Y^{3+}$ are similar in hardness, with $I_A$ values of 10.49 and 10.64, respectively, they differ greatly in pKa values (4.3 for $Sc^{3+}$ v. 7.7 for $Y^{3+}$)[1]. This manifests in the pH where hydrolysis begins for the two rare-earth cations, with $Sc^{3+}$ hydrolysis beginning at approximately pH 2.5 while $Y^{3+}$ hydrolysis occurs at neutral pH or above[27–29]. This latter characteristic is desirable for radiolabeling processes, which typically require pH values between 4 and 5, where hydrolysis products in a $Sc^{3+}$ system may limit the efficacy of a selected chelator[30]. Finally, the longer half-life of $^{86}$Y ($t_{1/2}$ = 14.74 h) covers a much longer portion (~2 days) of the pharmacokinetics of relevant therapeutic targeting vectors compared to either $^{68}$Ga ($t_{1/2}$ = 1.13 h) or $^{44}$Sc ($t_{1/2}$ = 3.97 h), which is a significant advantage as the latter PET isotopes are limited to measuring only early therapeutic uptake kinetics[31].

## Results and discussion

**Solution chemistry of rare earth complexes with 343-HOPO.** The UV-Visible absorbance spectra of 343-HOPO at pH 7.4 upon addition of $Sc^{3+}$ or $Y^{3+}$ are shown in Fig. 1. Compared to free

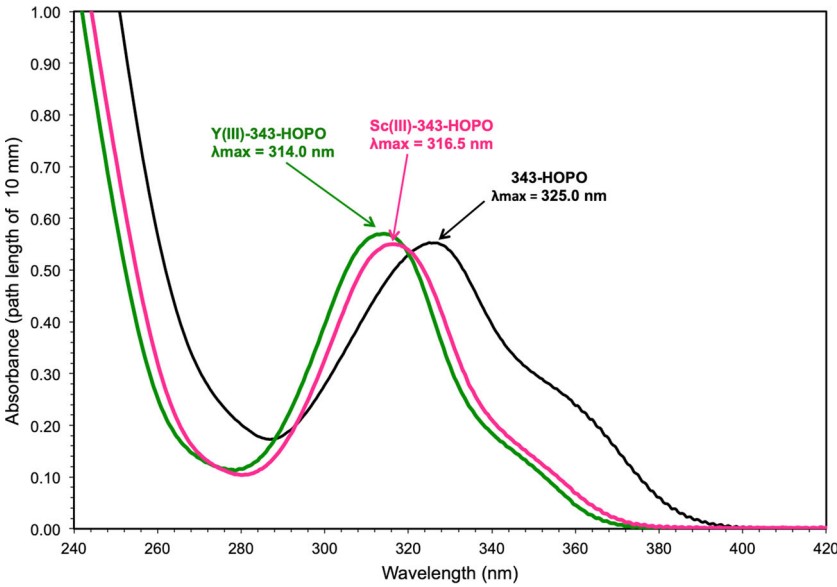

**Fig. 1 UV-Vis absorbance spectra of [Sc-343-HOPO]− and [Y-343-HOPO]−.** Samples contained 30 μM of 343-HOPO (black curve), 30 μM of 343-HOPO and 30 μM of $Sc^{3+}$ ions (pink curve), or 30 μM of 343-HOPO and 30 μM of $Y^{3+}$ ions (green curve). Path length: 10 mm. Background electrolyte: 25 mM HEPES buffer, pH = 7.4. Absorbance corrected from blank absorbance (25 mM HEPES). T = 25 °C.

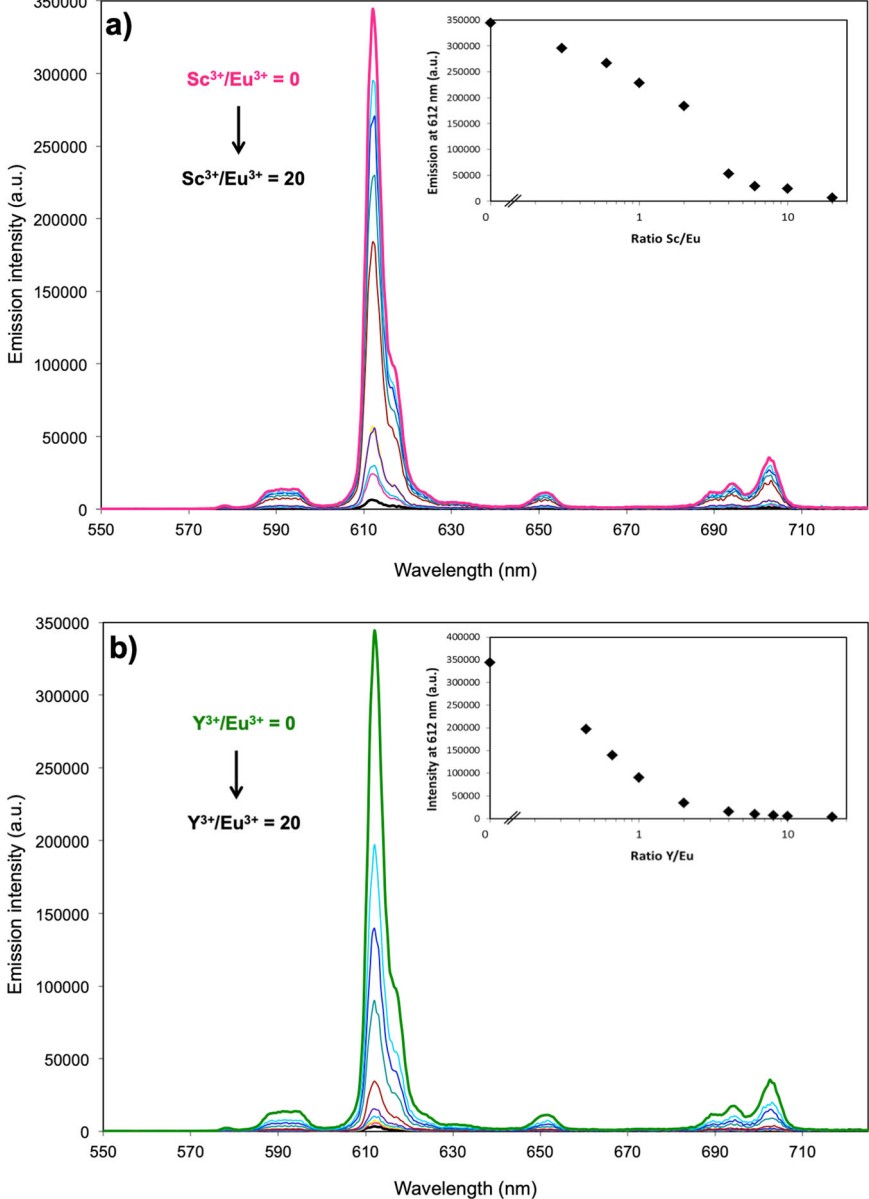

**Fig. 2 Spectrofluorimetric metal competition titrations.** [Eu-343-HOPO]$^-$ was competed against Sc$^{3+}$ (**a**) and Y$^{3+}$ (**b**). Insets: Emission intensity at 612 nm versus the ratio M/Eu (M = Sc$^{3+}$ or Y$^{3+}$). Ratio M/Eu = 0 to 20 equivalents. Background electrolyte: 2.5 mM HEPES + KCl. I = 0.1 M. pH = 7.4. T = 25°C.

343-HOPO, the UV spectrum is blue shifted upon addition of the metal ions Sc$^{3+}$ and Y$^{3+}$, which do not absorb in the highlighted spectral window. Changes in UV-Vis spectra are due to the binding of Sc$^{3+}$ or Y$^{3+}$ ions to the 1,2-HOPO groups of the chelator, and such results are consistent with those previously reported for the binding of lanthanide ions (Ln$^{3+}$, Ln = La$^{3+}$ to Lu$^{3+}$) to 343-HOPO and other 1,2-HOPO chelators[22,32,33]. The results depicted in Fig. 1 suggest that at physiological pH and at high metal concentrations, compared to radiopharmaceutical formulations, one equivalent of 343-HOPO is sufficient to form stable complexes with Sc$^{3+}$ or Y$^{3+}$ and prevent metal ion hydrolysis. Further confirmation of 343-HOPO complex stability can be seen in the $^{45}$Sc NMR spectrum (Supplementary Fig. 1) where the [Sc-343-HOPO]$^-$ chemical shift is observed downfield at ca. 62.5 ppm. The Sc$^{3+}$-aqua ion in dilute HCl produces a sharp signal at ca. 4 ppm (Supplementary Fig. 1), and the large $^{45}$Sc chemical shift upon complexation indicates 343-HOPO

effectively shields the $^{45}$Sc nucleus from solvent molecules (vide infra)[13]. The single peak in the $^{45}$Sc NMR spectrum for [Sc-343-HOPO]$^-$ also suggests no metal-centered isomerism on the NMR timescale, consistent with only a single species in solution.

Based on the solution thermodynamics of Ln$^{3+}$ complexes with 343-HOPO[22,34,35], the complexes that are likely formed in aqueous solutions with Sc$^{3+}$ and Y$^{3+}$ are: [M-343-HOPOH], [M-343-HOPO]$^-$, and [M-343-HOPO(OH)]$^{2-}$ (where M = Sc$^{3+}$ or Y$^{3+}$). The protonated and hydroxylated complexes, [M-343-HOPOH] and [M-343-HOPO(OH)]$^{2-}$, are only present in relatively narrow pH ranges for Ln$^{3+}$ cations (0 < pH < 3 and 9 < pH < 11, respectively), thus the most relevant complex at physiological pH (7.4) is [M-343-HOPO]$^-$. By analogy, the complexes [Sc-343-HOPO]$^-$ and [Y-343-HOPO]$^-$ are expected to be the predominant species in the pH range 3 to 9 (Supplementary Fig. 2). The proton-independent stability constants (log $\beta_{mlh}$) for both complexes were determined via spectrofluorimetric metal-competition titration

using the luminescent [Eu-343-HOPO]$^-$ complex as a reference[22]. Upon excitation at 325 nm, the europium complex exhibits an intense emission spectrum in the spectral window 500–800 nm. When another metal that competes with Eu$^{3+}$ for 343-HOPO binding is added to the system, the intensity of the emission spectrum of [Eu-343-HOPO]$^-$ decreases based on Eq. (1).

$$[Eu-343-HOPO]^-_{(aq)}+M^{3+}_{(aq)} = [M-343-HOPO]^-_{(aq)}+Eu^{3+}_{(aq)} \quad (1)$$

This equation can be applied if the competing M$^{3+}$ ions (here Sc$^{3+}$ and Y$^{3+}$) do not form a luminescent complex with 343-HOPO or generate an emission spectrum different from that of [Eu-343-HOPO]$^-$. 343-HOPO complexes of Sc$^{3+}$ and Y$^{3+}$ are not luminescent; consequently, upon addition of Sc$^{3+}$ and Y$^{3+}$ to a [Eu-343-HOPO]$^-$ solution, the emission intensity decreases until the concentration of [Eu-343-HOPO]$^-$ in the system is undetectable due to the formation of the spectroscopically silent [Sc-343-HOPO]$^-$ or [Y-343-HOPO]$^-$. Examples of this type of spectroscopic titration with both Sc$^{3+}$ and Y$^{3+}$ are shown in Fig. 2.

Utilizing the known metal hydrolysis constants, ligand protonation constants, and log $\beta_{110}$ value of [Eu-343-HOPO]$^{-}$[22,34], proton-independent stability constants values for [Sc-343-HOPO]$^-$ and [Y-343-HOPO]$^-$ were obtained using the method described above. Deconvolution of titration data were performed with *HypSpec*[34,36], which yielded log $\beta_{110}$ values of 25.16 ± 0.01 and 20.76 ± 0.09 for [Sc-343-HOPO]$^-$ and [Y-343-HOPO]$^-$, respectively. Notably, [Sc-343-HOPO]$^-$ is the most stable trivalent-343-HOPO complex reported to date, with a log $\beta_{110}$ value three orders of magnitude higher than its closest lanthanide or actinide analog (Table 1).

**Structural chemistry of rare earth complexes with 343-HOPO.** While the solution chemistry of 343-HOPO and its use in a variety of applications such as actinide decorporation and separations[37,38], post-MRI chelation therapy[39], or $^{89}$Zr PET imaging[40] is increasingly well-characterized, the structural chemistry of this chelator, with trivalent metals in particular, remains largely unexplored. Daumann et al. reported the first crystal structure featuring 343-HOPO and Eu$^{3+}$ in 2015[41], and this remained the only published structure featuring 343-HOPO and a trivalent

metal until this study. Our group has made progress recently on the structural chemistry of 343-HOPO with tetravalent p-block and d-block metals[42,43], and here we extend this knowledge to systems featuring Sc$^{3+}$ and Y$^{3+}$. Both complexes **1**, K[Sc(343-HOPO)]•DMF•(H$_2$O)$_x$, and **2**, K[Y(343-HOPO)]•DMF, crystallize in the space group P-1 upon slow evaporation from solutions of methanol containing 5% DMF. Complexes **1** and **2** both feature a single crystallographically unique rare-earth metal center with the Sc$^{3+}$ and Y$^{3+}$ cations adopting distorted square antiprismatic molecular geometries (Fig. 3). Sc$^{3+}$ and Y$^{3+}$ cations are each eight coordinate with all four 1,2-HOPO moieties of 343-HOPO binding in a bidentate manner and the average Sc$^{3+}$ and Y$^{3+}$–O bond distances are 2.213 Å and 2.347 Å, respectively. The asymmetric units for both complexes feature potassium cations for charge balancing purposes, as well as solvent molecules, which are significantly disordered for complex **1**. Akin to other examples of crystallographically characterized metal complexes of 343-HOPO, we find the Sc$^{3+}$ and Y$^{3+}$ complexes (**1** and **2**) display handedness, as racemic mixtures of Δ(λ) and Λ(δ) isomers. This notation, which we have previously employed, describes the configuration of the 1,2-HOPO moieties surrounding the metal center as either Δ or Λ, and the relative configuration of the butylene diamine backbone as either λ or δ (Supplementary Fig. 3)[43]. We have proposed that the extremely high thermodynamic stabilities of 343-HOPO complexes may allow for the chromatographic separation or resolution of these Δ(λ) and Λ(δ) enantiomers, and while outside the scope of the current study, we are currently working to achieve this goal and delineate differing in vivo behaviors that result due to complex handedness[43].

**343-HOPO complexation with $^{86}$Y and in vivo imaging experiments.** An advantage of working with 343-HOPO is that it has been evaluated for potential biomedical applications; thus, toxicology and pharmacology studies have already been completed[44–46]. In addition, the in vivo stability of metal complexes of 343-HOPO formed with both trivalent (Eu, Am, Cm) and tetravalent (Zr, Pu) metals has been investigated in multiple studies, using ex vivo radioanalytical techniques and in vivo PET imaging. All of these biodistribution studies have highlighted the unusual in vivo stability and quantitative, rapid hepatobiliary clearance of 343-HOPO complexes, as well as minimal or undetectable non-targeted uptake of metal ions, when compared to other commonly used chelators, including diethylenetriaminepentaacetic acid (DTPA) and desferrioxamine B (DFO)[22,37,40,47]. To assess the in vivo behavior and pharmacokinetics of the $^{86}$Y-343-HOPO complex, we administered the radiolabeled small molecule complex to young adult female Swiss Webster mice via intravenous injection. Mice were imaged at 15 min, 2 h, 24 h, and 48 h post-injection, and the results are highlighted in Fig. 4, Supplementary Fig. 4, and Supplementary Table 4, largely confirming biodistribution patterns previously noted with other metal ions. At 15 min, the vast majority of the radioactivity is observed in the gall bladder and gastrointestinal tract, indicating hepatic clearance is the main excretion pathway for $^{86}$Y-343-HOPO, with a small amount of activity appearing in the bladder, suggesting a minor renal clearance pathway as well. This is confirmed at 2 h when the majority of remaining radioactivity is found in the gastrointestinal tract and the liver. At 24 and 48 h, very little detectable radioactivity remains (Fig. 4, Supplementary Fig. 4, and Supplementary Table 4), which suggests the pharmacokinetics of the $^{86}$Y-343-HOPO complex are rapid. PET images at all four time points also display minimal kidney uptake, an important feature that may prevent dose-limiting nephrotoxicity (Fig. 4, Supplementary Fig. 4, and Supplementary Table 4), in contrast to $^{86}$Y complexes formed with DOTA or DTPA[48]. Moreover, the

**Table 1 Summary of proton-independent stability constants with 343-HOPO and trivalent metal cations.**

| Cation | log $\beta_{110}$ | Reference(s) |
|---|---|---|
| Sc$^{3+}$ | 25.16 ± 0.01 | this work |
| Y$^{3+}$ | 20.76 ± 0.09 | this work |
| La$^{3+}$ | 16.4(3) | 22 |
| Ce$^{3+}$ | 17.4(5) | 35 |
| Pr$^{3+}$ | 18.2(4) | 22 |
| Nd$^{3+}$ | 18.7(1) | 22 |
| Sm$^{3+}$ | 19.7(3) | 22 |
| Eu$^{3+}$ | 20.2(2) | 22, 34 |
| Gd$^{3+}$ | 20.5(1) | 22 |
| Tb$^{3+}$ | 20.9(1) | 22 |
| Dy$^{3+}$ | 21.2(1) | 22 |
| Ho$^{3+}$ | 21.5(1) | 22 |
| Er$^{3+}$ | 21.7(1) | 22 |
| Tm$^{3+}$ | 22.0(1) | 22 |
| Yb$^{3+}$ | 22.2(1) | 22 |
| Lu$^{3+}$ | 21.2(1) | 22 |
| Am$^{3+}$ | 20.4(2) | 54 |
| Cm$^{3+}$ | 21.8(4) | 47 |

Log $\beta_{110}$ values, at 25°C and I = 0.1 M, either experimentally determined in this work or previously reported[22,34,35,47,54].

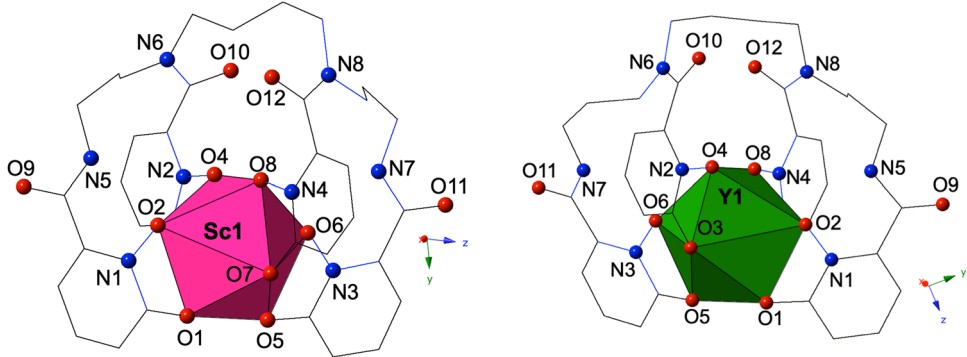

**Fig. 3 Solid-state structures of Sc³⁺-343-HOPO and Y³⁺-343-HOPO complexes.** Polyhedral representations of complexes **1** and **2**, where pink and green polyhedra are Sc³⁺ and Y³⁺ centers, respectively, and spheres represent oxygen atoms (red) and nitrogen atoms (blue). Hydrogen atoms, potassium cations, and solvent molecules are omitted for clarity.

**Fig. 4 Coronal PET images of ⁸⁶Y-343-HOPO.** Three healthy mice were administered ⁸⁶Y-343-HOPO (93.1 μCi [3.44 MBq] in 10x PBS) via tail vein injection and imaged between 15 min and 24 h after injection. The gall bladder (a), the gastrointestinal tract (b), and the bladder (c) can be visualized at the 15 min timepoint. The ⁸⁶Y-343-HOPO complex primarily undergoes rapid hepatic clearance and no uptake of ⁸⁶Y in the skeleton is observed.

lack of radioactivity accumulating in the skeleton or other organs suggests that the ⁸⁶Y complex remains intact over several hours in vivo[12], thereby demonstrating that 343-HOPO is well suited for targeted in vivo applications with yttrium radioisotopes.

Herein, we characterized the complexation of Sc³⁺ and Y³⁺ with 343-HOPO in solution via UV-Vis spectrophotometry, NMR spectroscopy, and spectrofluorimetric metal-competition titrations, and in the solid-state via single crystal X-ray diffraction. We

experimentally observed high thermodynamic stability for the binding of 343-HOPO with Sc³⁺ and Y³⁺, and crystallographic analysis agrees with solution-state NMR results, indicating rare-earth cations are well shielded from solvent molecules. Proof of concept ⁸⁶Y-343-HOPO PET imaging experiments revealed hepatic clearance, rapid pharmacokinetics, and no detectable radioactivity accumulating in bones or other organs, all of which are valuable for future work pairing ⁸⁶Y with therapeutic isotopes

of interest. In combination, these results indicate that 343-HOPO is likely a promising chelator for both the $^{44/47}$Sc and $^{86/90}$Y theranostic isotope pairs. Future studies will focus on expanding efforts to 343-HOPO conjugates[49] and siderocalin fusion proteins[50,51] using Sc$^{3+}$ and Y$^{3+}$, as well as lanthanides and actinides of theranostic interest.

## Methods

**Incremental spectrofluorimetric titrations.** The spectrofluorimetric metal-metal competition titration method developed by Sturzbecher-Hoehne et al.[22] was used to evaluate the thermodynamic stability of the Sc(III)-343-HOPO and Y(III)-343-HOPO complexes. This method used the luminescent [Eu-343-HOPO]$^-$ complex as a reference as this complex has been thoroughly characterized[34,41], and typically, 10–12 samples of 2 mL were prepared per titration. Each sample contained 0.1 μM Eu(III), 0.1 μM 343-HOPO, and 0 to 20 equivalents of competing metal ions (here Sc$^{3+}$ or Y$^{3+}$). Samples were buffered with 0.1 M HEPES at pH 7.4, and after 24 h of equilibration in a thermostated bath at 25 °C, the emission spectra of the samples were acquired with a HORIBA Jobin Yvon IBH FluoroLog-3 spectrofluorimeter in steady state mode. Emission (550–725 nm–350 data points) was monitored perpendicular to the excitation pulse after excitation of the sample at 325 nm. Slits were set at 2.0 nm and 1.0 nm for the emission and excitation monochromators, respectively, and the integration time was 0.4 s per point.

**343-HOPO radiolabeling with $^{86}$Y.** A 2.86 mL 10× phosphate buffered saline (PBS) solution containing 5 μL of 343-HOPO (34.9 μM) in DMSO was added to the $^{86}$Y stock solution in 0.1 M HCl. To minimize radioactivity dose, the solution was not shaken and was instead allowed to equilibrate at room temperature inside a lead pig for 10 min. $^{86}$Y-343-HOPO binding was confirmed via a wild-type siderocalin (Scn) binding assay developed in-house, wherein an aliquot of $^{86}$Y-343-HOPO complex was combined with Scn, and upon Scn recognition, separation from free $^{86}$Y was done via spin filtration[52]. This technique has been established for charge-based separation of neighboring actinides, and was adapted here to qualitatively confirm $^{86}$Y complexation by 343-HOPO, with characterization done using an optimized Ludlum 2224-1 Alpha-Beta Scaler-Ratemeter[53]. An aliquot was also taken to determine the activity via γ-spectroscopy, wherein the 307.00, 443.13, 580.57, and 777.37 keV gamma lines of $^{86}$Y were measured on a P-Type High Purity Germanium γ-Spectrometer.

**Small animal PET imaging.** All procedures and protocols used in small animal PET imaging studies were reviewed and approved by the Institutional Animal Care and Use Committee at Lawrence Berkeley National Laboratory. Experiments were performed in compliance with guidelines from the Association for Assessment and Accreditation of Laboratory Animal Care International (AAALAC) in AAALAC accredited facilities. The animals used were healthy young adult (11–12 weeks old) female (32.6 ± 1.5 g) Swiss Webster mice (Simonsen Laboratories, Gilroy, CA, USA), which were given water and food ad libitum, and kept under a 12-h light cycle with controlled temperature (18–22 °C) and relative humidity (30–70%). Intravenous (iv) injections into a warmed lateral tail vein were performed under isoflurane anesthesia. Three mice were injected iv with a single 200 μL dose of $^{86}$Y-343-HOPO (3.44 MBq, 93.1 μCi, 2.18 nM), the preparation of which is described above. The mice were imaged at 15 min, 2 h, 24 h, and 48 h after injection on a Concorde microPET R4 which supports a transaxial resolution of 1.66 mm FWHM, in the head first supine position. During the scan, mice were anesthetized with a mixture of isoflurane and oxygen. An energy window of 350 − 650 keV and a coincidence timing window of 6 ns were used during image acquisition. Mice were subsequently euthanized under isoflurane anesthesia via cervical dislocation following the 48-h time point.

**Reporting summary.** Further information on research design is available in the Nature Research Reporting Summary linked to this article.

## Data availability

Additional experimental details (see Supplementary Methods), as well as X-ray crystallographic files in CIF format, ORTEP figures of both complexes, and additional figures and tables are included in the supplementary information. All data generated or analyzed during this study are included in this published article (or in the supplementary information). CCDC 1983231 and 1983232 contain the supplementary crystallographic information for each complex, which can be obtained free of charge from The Cambridge Crystallographic Data Center. The supplementary crystallographic information is also available as Supplementary Data 1 and 2.

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

## Acknowledgements

This work was supported by the U.S. Department of Energy (DOE), Office of Science, Office of Basic Energy Sciences, Chemical Sciences, Geosciences, and Biosciences Division at the Lawrence Berkeley National Laboratory under Contract DE-AC02-05CH11231. We acknowledge additional support from U.S. DOE Integrated University Program graduate research fellowships (T.D.L. and K.M.S.) and a Nuclear Regulatory Commission Faculty Development Grant (NRC-HQ-84-14-G-0052; R.J.A.). We gratefully recognize the U.S. DOE, Office of Science, National Isotope Development Center (NIDC) subprogram within the Office of Nuclear Physics for supplying [86]Y. Use of the Advanced Light Source (A.L.S.) is supported by the U.S. DOE, Office of Science, Office of Basic Energy Sciences, under Contract DE-AC02-05CH11231. We thank Dr. Simon J. Teat for training and guidance throughout our crystallography experiments at the A.L.S.

## Author contributions

All work was performed at the Lawrence Berkeley National Laboratory. K.P.C., G.J.-P.D., and R.J.A. designed the research. G.J.-P.D. and W.W.L. performed solution state experiments. K.P.C. and G.J.-P.D. grew crystals for X-ray diffraction (XRD) experiments. K.P.C. and T.D.L. collected and analyzed XRD data. K.P.C. and K.M.S. prepared the radiolabeled [86]Y complex. T.A.B. and D.D.A. performed in vivo PET experiments. All authors discussed the experimental results and contributed to the manuscript.

## Competing interests

The authors declare no competing interests.
