## [Peer Review File · Communications Chemistry]

Reviewers' comments:

Reviewer #1 (Remarks to the Author):

The manuscript entitled "Advancing $^{44}\text{Sc}/^{47}\text{Sc}$ and $^{86}\text{Y}/^{90}\text{Y}$ Theranostic Radiopharmaceuticals through Fundamental Coordination Chemistry" by Carter et al outlines the investigation of the octadentate siderophore 3,4,3-LI(1,2-HOPO) for the chelation of scandium and yttrium as well as their corresponding radioactive isotopes. Chelation approaches to these metal ions are currently critically needed in order to enable development of new and improved radiopharmaceuticals. The manuscript outlines some promising preliminary results, however, additional data is required to adequately support conclusions made.

- Title: It would be prudent to revise the title to not include the listed isotopes, as no $^{44}\text{Sc}/^{47}\text{Sc}$ or ^{90}Y radiochemistry has been carried out to date.

- Thermodynamic stability: please include pH dependent speciation diagrams for both complex formations. This is especially critical as HOPO may form protonated complex species at biologically relevant pH values, which may indicate an altered coordination complex in solution when compared to the solid-state structures.

- Please include ^1H spectral data on the coordination complexes. Also, please contextualize the obtained ^{45}Sc NMR data with published literature. For instance, the chemical shift (62.5 ppm) reported here could indicate a lower CN than 8, which is especially plausible if the ligand is partially protonated.

- Data on the animal studies must include biodistribution results. PET images, especially without CT co-registration) are not sufficiently informative to indicate in vivo stability. Where does free ^{86}Y accumulate? Radiochemical labeling experiments are also not sufficiently detailed. Please include data and results on the corresponding radiochemical characterization. Was size exclusion chromatography used to characterize the Sc^{III} binding?

Once the items above have been appropriately addressed/included, the manuscript can be reconsidered for publication.

Reviewer #2 (Remarks to the Author):

The manuscript describes the results of the fundamental coordination chemistry and potential therapeutic and theranostic applications of a chelating ligand, 3,4,3-HOPO, for its high affinity for trivalent lanthanide cations. The authors show that 3,4,3-HOPO shows high thermodynamic stability and fast radiolabeling kinetics for the Sc^{3+} and Y^{3+} via spectrofluorimetric metal-competition titrations, single crystal X-ray diffraction and PET imaging, indicating that 3,4,3-HOPO is a promising chelator for in vivo applications with both metals. Overall, this work is nicely done, but with some issues that need to be addressed before publication.

1. The authors claim that ^{68}Ga as the PET imaging agent mimic of ^{177}Lu or ^{90}Y for β^- therapy is improper for the different ionic radii, coordination chemistry and in vivo behavior, and $^{44}\text{Sc}/^{47}\text{Sc}$ and $^{86}\text{Y}/^{90}\text{Y}$ was considered as great promise in this arena. Please give more information about the difference of ionic radii, coordination chemistry and in vivo behavior. In addition, it is confused that Sc is proper for β^- therapy, please provide more information to prove it.

2. "Both ^{44}Sc and ^{86}Y offer several advantages compared to the commonly utilized ^{68}Ga , including longer half-lives (3.97 and 14.74 hours, respectively) and preferences for higher coordination numbers" in page 2 paragraph 2, please provide a further explanations on higher coordination number is an advantage for β^- therapy.

3. The authors declared that Sc^{3+} and Y^{3+} binding with 3,4,3-HOPO was found to be robust, with

both high thermodynamic stability and fast radiolabeling kinetics. I do not see the data supporting the fast kinetics of the binding behavior.

4. ESI-MS spectrometry is useful for confirmation the formation of 1:1 metal-ligand complex of [M-343-HOPO]⁻ in solution state.

5. In this work, the authors analysis the disadvantages of DOTA as chelator for its poor binding affinity and low selectivity. Please provide more discussion on the direct comparison on the formation constants between the two ligands.

6. Is it proper to use the Eu³⁺ as the competitor in the titration experiments of Y-343-HOPO for the similar formation constants?

7. Small animal PET showed obvious thyroid uptake in 15 min and 2 hours. The author need to provide quantitative analysis of the PET imaging or biodistribution. The organs should be marked on the figures. Moreover, the author should prove the radio-stability of ⁸⁶Y-343-HOPO in PBS and serum. Also, a positive control group of DOTA are suggested to be performed in the PET imaging to better understand the advantage of 343-HOPO.

Reviewer #3 (Remarks to the Author):

This manuscript provides considerable information regarding the basic coordination chemistry of 343-HOPO with Sc(III) and Y(III) with the purpose of using this chemistry to move this chelating agent forward into use with radionuclides of medical interest for imaging and therapy. Justification and rationale for this is based principally in the deficit of DOTA being limited by slow formation kinetics to form radiometal complexes at room temperature. Traversing this limitation is something useful.

There are however a number of issues that need to be resolved.

Page 2 – overlooks the potential of using Lu-177 itself for imaging – gamma and SPECT imaging is possible so this argument of a mis-match with Ga-68 both in chemistry and half-life might not be as significant.

Page 3 – this transmetallation issue is really not real if the complex is stable, particularly so with DOTA which tends to form kinetically inert complexes with most of the radionuclides of interest – leakage or transmetallation tends to be the result of poor choices a mis-match if chemistry, or incompletely formed complexes which can occur at room temperature with DOTA. This broad statement needs to be revised.

Hydroxypyridinone moieties – these are quite different donor groups from polyaminocarboxylate ligands and while they may form good complexes, they are also quite different in lipophilicity and also solubility. That detail needs to be noted and addressed – more later on this point....

Page 10 – hepatic clearance – this is where that lipophilicity comes into play –

polyaminocarboxylate complexes are almost entirely renal clearance agents and exceedingly rapid – 95%+ cleared within 24 h. This needs to be noted and discussed as well as what impact this can have on the biodistribution of targeting agents once this HOPO is conjugated to them – such effects have been noted in the past so this is something that should be expected. “suggests the pharmacokinetics” -- “suggests the clearance pharmacokinetics” - really not looking at the entire full detailed PK. PET imaging PK really is only assessing the complex’s properties and it’s too rapid to really test real in vivo stability as needed once this HOPO is conjugated to targeting agents – cannot extrapolate to anything beyond just the complex itself at this stage with this data. Maybe lower renal tox but side-stepped hepatic route tox. Re-routing the clearance may not eliminate tox particularly when the route is actually a bit slower. Statements regarding skeleton and other tissue uptake, etc. cannot be made with real accuracy based on just PET imaging. A full detailed biodistribution animal study is required to define that status. Lastly, probably should have also looked to see just what was excreted and what form it was in to define and confirm that the complex remained intact – simple metabolic study by HPLC analysis would work.

Page 12 – conclusion is overstated for what is really an incomplete preliminary study – really have not done the definitive study even at this stage to demonstrate basic stability in vivo of the

complexes of either Sc(III) or Y(III) radionuclides. That will have to be done as well as done again once this agent is conjugated or incorporated into any targeting agent.

Rebuttal Letter to Reviewers

April 3, 2020

We appreciate the reading of this paper by the reviewers and have made changes to the manuscript accordingly, as described in the following pages.

We hope that the manuscript is now acceptable for publication in Communications Chemistry.

Sincerely,

Rebecca Abergel.

Referee 1

Comments:

The manuscript entitled "Advancing $^{44}\text{Sc}/^{47}\text{Sc}$ and $^{86}\text{Y}/^{90}\text{Y}$ Theranostic Radiopharmaceuticals through Fundamental Coordination Chemistry" by Carter et al outlines the investigation of the octadentate siderophore 3,4,3-LI(1,2-HOPO) for the chelation of scandium and yttrium as well as their corresponding radioactive isotopes. Chelation approaches to these metal ions are currently critically needed in order to enable development of new and improved radiopharmaceuticals. The manuscript outlines some promising preliminary results, however, additional data is required to adequately support conclusions made.

Title: It would be prudent to revise the title to not include the listed isotopes, as no Sc-44/47 or Y-90 radiochemistry has been carried out to date.

Our Response: We appreciate the feedback and have updated the title per the referee's suggestion.

Action Taken: Title has been updated to 'Developing Scandium and Yttrium Coordination Chemistry to Advance Theranostic Radiopharmaceuticals'.

- Thermodynamic stability: please include pH dependent speciation diagrams for both complex formations. This is especially critical as HOPO may form protonated complex species at biologically relevant pH values, which may indicate an altered coordination complex in solution when compared to the solid-state structures.

Our Response: Previous studies looking at the complexation of HOPO with trivalent lanthanides, actinides, and transition metals (Sturzbecher-Hoehne *et al.* Dalton Trans. 2011, 40, 8340-8346; Deblonde *et al.* Inorg. Chem. 2013, 52, 8805-8811; Sturzbecher-Hoehne *et al.* Chem. Euro. J. 2014, 20, 9962-9968; Allred *et al.* Proc. Natl. Acad. Sci. 2015, 112, 10342-10347; Sturzbecher-Hoehne *et al.* Dalton Trans. 2016, 45, 9912-9919; Kelley *et al.* Inorg. Chem. 2018, 57, 5352-5363; Pallares *et al.* Inorg. Chem. 2020, 59, 2030-2036) have demonstrated that HOPO does not form a protonated complex at biologically relevant pH values; however, for completeness we have prepared speciation diagrams for Sc^{3+} and Y^{3+} systems.

Action Taken: Speciation diagrams for Sc^{3+} and Y^{3+} with 343-HOPO have been added in Figure S2 of the Supporting Information and are referred to on page 5 of the manuscript.

- Please include ^1H spectral data on the coordination complexes. Also, please contextualize the obtained Sc-45 NMR data with published literature. For instance, the chemical shift (62.5 ppm) reported here could indicate a lower CN than 8, which is especially plausible if the ligand is partially protonated.

Our Response: As ^1H NMR would not provide us information that we did not already know from UV-vis spectra or spectrofluorimetric metal-competition titrations, these experiments had not initially been conducted, and due to our lab being closed during the Covid-19 outbreak, this is not data we can obtain in the near future. We do find merit in the suggestion to add context to ^{45}Sc NMR results and have taken this chance to do so. The speciation diagrams we added to the Supporting Information (detailed above) definitively indicate the ligand is fully deprotonated at the pH where ^{45}Sc NMR was collected.

Action Taken: ^{45}Sc NMR results section on pgs. 4 and 5 of the manuscript has been updated as follows, "Further confirmation of 343-HOPO complex stability can be seen in the ^{45}Sc NMR spectrum (Figure S1, Supporting Information) where the $[\text{Sc}-343\text{-HOPO}]^-$ chemical shift is observed downfield at ca. 62.5 ppm. The Sc^{3+} -aqua ion in dilute HCl produces a sharp signal at ca. 4 ppm (Figure S1,

Supporting Information), and the large ^{45}Sc chemical shift upon complexation indicates 343-HOPO effectively shields the ^{45}Sc nucleus from solvent molecules (*vide infra*).¹³ The single peak in ^{45}Sc NMR spectrum for [Sc-343-HOPO] also suggests no metal-centered isomerism on the NMR timescale, consistent with only a single species in solution.”

- Data on the animal studies must include biodistribution results. PET images, especially without CT co-registration) are not sufficiently informative to indicate *in vivo* stability. Where does free Y-86 accumulate? Radiochemical labeling experiments are also not sufficiently detailed. Please include data and results on the corresponding radiochemical characterization. Was size exclusion chromatography used to characterize the Scn binding?

Our Response: We must respectfully push back a little as insisting on biodistribution results does change the scope and nature of this paper. Moreover, obtaining biodistribution data is unfortunately not possible as our lab is closed until further notice, as a result of Covid-19. Previous biodistribution studies of M(III)-343-HOPO complexes have all indicated complexes are stable and clear hepatically and quantitatively within 24 hours. (See: Eu(III)--Sturzbecher-Hoehne *et al.* Dalton Trans. 2011, 40, 8340-8346, Am(III)--Kullgren *et al.* Toxicol. Mech. Meth. 2013, 23, 18-26, and Cm(III)--Sturzbecher-Hoehne *et al.* Chem. Eur. J. 2014, 20, 9962-9968)

Free ^{86}Y accumulates primarily in the skeleton with a very small dose to the liver (Herzog *et al.* J Nucl. Med. 1993, 34, 2222-2226 (ref. 12 in the manuscript)), which PET images would reveal even in the absence of co-registered CT results.

Regarding radiochemical labeling, we have taken this opportunity to add additional information on the siderocalin assay used to confirm ^{86}Y complexation, and this information has been added to the supporting information.

Actions taken:

We have added context and several references in the section describing the *in vivo* imaging experiments on page 10 of the manuscript: “In addition, the *in vivo* stability of metal complexes of 343-HOPO formed with both trivalent (Eu, Am, Cm) and tetravalent (Zr, Pu) metals has been investigated in multiple studies, using *ex vivo* radioanalytical techniques and *in vivo* PET imaging. All of these biodistribution studies have highlighted the unusual *in vivo* stability and quantitative, rapid hepatobiliary clearance of 343-HOPO complexes, as well as minimal or undetectable non-targeted uptake of metal ions, when compared to other commonly used chelators, including diethylenetriaminepentaacetic acid (DTPA) and desferrioxamine B (DFO).^{22,37,39,42}”

In ‘343-HOPO Radiolabeling’ section of Supporting Information the following sentences have been added: “ ^{86}Y -343-HOPO binding was confirmed via a wild-type siderocalin (Scn) binding assay developed in-house, wherein an aliquot of ^{86}Y -343-HOPO complex was combined with Scn, and upon Scn recognition, separation from free ^{86}Y was done via spin filtration.¹² This technique has been established for charge-based separation of neighboring actinides, and was adapted here to qualitatively confirm ^{86}Y was successfully complexed by 343-HOPO, with characterization done via an optimized Ludlum 2224-1 Alpha-Beta Scaler-Ratemeter.¹³”

Referee 2

Comments:

The manuscript describes the results of the fundamental coordination chemistry and potential therapeutic and theranostic applications of a chelating ligand, 343-HOPO, for its high affinity for trivalent lanthanide cations. The authors show that 343-HOPO shows high thermodynamic stability

and fast radiolabeling kinetics for the Sc^{3+} and Y^{3+} via spectrofluorimetric metal-competition titrations, single crystal X-ray diffraction and PET imaging, indicating that 343-HOPO is a promising chelator for in vivo applications with both metals. Overall, this work is nicely done, but with some issues that need to be addressed before publication.

1. The authors claim that ^{68}Ga as the PET imaging agent mimic of ^{177}Lu or ^{90}Y for β^- therapy is improper for the different ionic radii, coordination chemistry and in vivo behavior, and $^{44}\text{Sc}/^{47}\text{Sc}$ and $^{86}\text{Y}/^{90}\text{Y}$ was considered as great promise in this arena. Please give more information about the difference of ionic radii, coordination chemistry and in vivo behavior. In addition, it is confused that Sc is proper for β^- therapy, please provide more information to prove it.

Our Response: We have provided the additional details requested by the referee. The origin of the coordination chemistry differences largely stems from Ga^{3+} predominantly forming hexadentate complexes, while Lu^{3+} , Y^{3+} , and Sc^{3+} all prefer octadentate coordination. Changes in in-vivo behavior as a result of coordination chemistry mismatches between Ga^{3+} and the rare earth cations of therapeutic interest have been detailed in refs. 9-11. ^{47}Sc is the scandium isotope with potential for use in β^- therapy. We mention it here to demonstrate the potential of theranostic applications with both Sc^{3+} and Y^{3+} even though Sc work is underdeveloped due to the limited availability of ^{47}Sc .

Action Taken: Penultimate sentence of first paragraph of introduction has been updated to read, “However, this approach is hindered by the different ionic radii of these elements (CN=6, 0.62 Å for Ga^{3+} ; CN=8, 0.977 Å for Lu^{3+} , CN=8, 1.019 Å for Y^{3+}),^{7,8} which leads to mismatches in coordination chemistry and varying *in vivo* behavior.⁹⁻¹¹”

2. “Both ^{44}Sc and ^{86}Y offer several advantages compared to the commonly utilized ^{68}Ga , including longer half-lives (3.97 and 14.74 hours, respectively) and preferences for higher coordination numbers” in page 2 paragraph 2, please provide a further explanations on higher coordination number is an advantage for β^- therapy.

Our Response: We appreciate this comment as it provides an opportunity to add clarity to the manuscript. Higher coordination numbers are not necessarily an advantage for β^- therapy; however, they are beneficial when you are pairing an imaging isotope with a therapeutic isotope (i.e. ^{86}Y with ^{177}Lu is a better coordination chemistry match than ^{68}Ga with ^{177}Lu).

Action Taken: First sentence of second paragraph in introduction has been updated to read, “Both ^{44}Sc and ^{86}Y offer several advantages compared to the commonly utilized ^{68}Ga as diagnostic pairs for ^{177}Lu or ^{90}Y , including longer half-lives (3.97 and 14.74 hours, respectively), higher resolution PET images (Table S2, Supporting Information), and preferences for higher coordination numbers.^{15,16}”

3. The authors declared that Sc^{3+} and Y^{3+} binding with 343-HOPO was found to be robust, with both high thermodynamic stability and fast radiolabeling kinetics. I do not see the data supporting the fast kinetics of the binding behavior.

Our Response: We used the term fast radiolabeling kinetics to describe the radiolabeling procedure for ^{86}Y -343-HOPO, which was done at room temperature in less than ten minutes. We acknowledge that kinetics experiments were not conducted, and as a result we aimed to be measured in our language. While we feel fast radiolabeling kinetics is an appropriate description of the Sc^{3+} and Y^{3+} binding process, we have adjusted our description of rare-earth binding kinetics in-line with the referee’s suggestion.

Action Taken: Sentence in abstract featuring “fast radiolabeling kinetics” has been changed to read, “ Sc^{3+} and Y^{3+} binding with 343-HOPO was found to be robust, with both high thermodynamic stability

and fast room temperature radiolabeling, indicating that 343-HOPO is a promising chelator for *in vivo* applications with both metals.”

4. ESI-MS spectrometry is useful for confirmation the formation of 1:1 metal-ligand complex of [M-343-HOPO]⁻ in solution state.

Our Response: We agree, and such work has been done previously in our group on the entire lanthanide-343-HOPO series (see Table S2 in Sturzbecher-Hoehne *et al.* Dalton Trans. 2011, 40, 8340-8346). Given the similarities in coordination chemistry between Sc³⁺ and Y³⁺ and lanthanide(III) cations, we did not feel ESI-MS on Sc- and Y-343-HOPO complexes would add more to the manuscript, and thus data has not been collected. Moreover, due to the Covid-19 shutdown of our lab, such data cannot currently be collected.

Action Taken: None

5. In this work, the authors analysis the disadvantages of DOTA as chelator for its poor binding affinity and low selectivity. Please provide more discussion on the direct comparison on the formation constants between the two ligands.

Our Response: We have clarified some of our discussions about DOTA to more explicitly highlight the limitations in using it as a radiotherapeutic chelator in-line with this suggestion and those of the other referees. We do not mention DOTA binding affinity as the thermodynamic binding constants for DOTA and 343-HOPO are similar (log β values of 24.5 and 20.76 for DOTA and 343-HOPO, respectively) (DOTA log β value is from Kumar *et al.* Inorg. Chem. 1994, 33, 3567-3575). The difference in the two chelators is the conditions used for radiolabeling as 343-HOPO provides similarly robust (thermodynamically) binding at room temperature in less than ten minutes. The common acyclic chelator diethylenetriaminepentaacetic acid (DTPA) can also radiolabel ⁸⁶Y at room temperature (see Le Fur *et al.* Angew. Chem. Int. Ed. 2020, 59, 1474-1478); however, this complex is not kinetically inert, unlike the complex formed between yttrium (or scandium) and 343-HOPO.

Action Taken: None here but see adjustments made related to DOTA in further discussion, in line with the next referee suggestions.

6. Is it proper to use the Eu³⁺ as the competitor in the titration experiments of Y-343-HOPO for the similar formation constants?

Our Response: Yes, Eu³⁺ is a model competitor for Y³⁺ in a spectrofluorimetric metal-competition titration. The similar formation constants of the two metal cations means less Y³⁺ is required for displacement of Eu³⁺ from 343-HOPO binding. Moreover, given the bright luminescence of Eu-343-HOPO (see Abergel *et al.* Inorg. Chem. 2009, 48, 10868-10870), it is a system that is very sensitive to changes, which is ideal for extracting thermodynamic data.

Action Taken: None

7. Small animal PET showed obvious thyroid uptake in 15 min and 2 hours. The author need to provide quantitative analysis of the PET imaging or biodistribution. The organs should be marked on the figures. Moreover, the author should prove the radio-stability of ⁸⁶Y-343-HOPO in PBS and serum. Also, a positive control group of DOTA are suggested to be performed in the PET imaging to better understand the advantage of 343-HOPO.

Our Response: We must respectfully push back on the characterization of “obvious” thyroid uptake; however, the referee makes a number of good suggestions that we have adopted. Organs have now

been marked in Figure 4 of the manuscript and quantitative analysis of the PET images has been done via mean standardized uptake value (SUV) calculations within a region of interest (ROI), which are included in Table S4 of the Supporting Information.

^{86}Y -343-HOPO is known to be stable in PBS and was prepared in this matrix for PET experiments herein. While serum stability studies are worthwhile, we believe them to be beyond the scope of this work.

^{86}Y -DOTA PET was recently done by Le Fur *et al.* (Angew. Chem. Int. Ed. 2020, 59, 1474-1478) so we have put our results in context with this recent work.

Action Taken: Figure 4 has been updated to include organ labels and quantitative analysis of PET images has been done via SUV calculations (included in Table S4, Supporting Information). Comparison between ^{86}Y -343-HOPO results and those with DOTA are included in the following sentence beginning at the bottom of pg. 10 of manuscript: “At 24 and 48 hours, very little detectable radioactivity remains (Figures 4, S4, and Table S4 (Supporting Information)), which suggests the pharmacokinetics of the ^{86}Y -343-HOPO complex are rapid. PET images at all four time points also display minimal kidney uptake, an important feature that may prevent dose-limiting nephrotoxicity (Figures 4, S4, and Table S4 (Supporting Information)), in contrast to ^{86}Y complexes formed with DOTA or the well-known chelator diethylenetriaminepentaacetic acid (DTPA).⁴⁹” (where ref. 49 is Le Fur *et al.* Angew. Chem. Int. Ed. 2020, 59, 1474-1478)

Referee 3

This manuscript provides considerable information regarding the basic coordination chemistry of 343-HOPO with Sc(III) and Y(III) with the purpose of using this chemistry to move this chelating agent forward into use with radionuclides of medical interest for imaging and therapy. Justification and rationale for this is based principally in the deficit of DOTA being limited by slow formation kinetics to form radiometal complexes at room temperature. Traversing this limitation is something useful.

There are however a number of issues that need to be resolved.

Page 2 – overlooks the potential of using Lu-177 itself for imaging – gamma and SPECT imaging is possible so this argument of a mis-match with Ga-68 both in chemistry and half-life might not be as significant.

Our Response: While we agree that ^{177}Lu can be used for SPECT imaging, PET and SPECT imaging do not offer equivalent information. The sensitivity and resolution of PET images are known to be superior to SPECT (see Rahmim and Zaidi Nucl Med. Comm. 2008, 29, 193-207), and thus we view it as valuable and relevant to develop PET isotope coordination chemistry to pair with therapeutic isotopes such as ^{177}Lu or ^{90}Y .

Action Taken: None

Page 3 – this transmetallation issue is really not real if the complex is stable, particularly so with DOTA which tends to form kinetically inert complexes with most of the radionuclides of interest – leakage or transmetallation tends to be the result of poor choices a mis-match if chemistry, or incompletely formed complexes which can occur at room temperature with DOTA. This broad statement needs to be revised.

Our Response: This is a fair point and one that distracts from the main limitation of DOTA as a radiotherapeutic chelator (i.e. poor radiolabeling kinetics that require high temperatures and long

heating times to ensure metal complexation). As a result we have removed this sentence from the introduction of the manuscript.

Action Taken: Deleted the following sentence from the introduction section of the manuscript, “Moreover, DOTA is known to be a non-specific chelator,¹⁹ so transmetalation by endogenous metals is a constant concern for DOTA-based chelates and pharmaceuticals.²⁰”

Hydroxypyridinone moieties – these are quite different donor groups from polyaminocarboxylate ligands and while they may form good complexes, they are also quite different in lipophilicity and also solubility. That detail needs to be noted and addressed – more later on this point....

Page 10 – hepatic clearance – this is where that lipophilicity comes into play – polyaminocarboxylate complexes are almost entirely renal clearance agents and exceedingly rapid – 95%+ cleared within 24 h. This needs to be noted and discussed as well as what impact this can have on the biodistribution of targeting agents once this HOPO is conjugated to them – such effects have been noted in the past so this is something that should be expected. “suggests the pharmacokinetics” -- “suggests the clearance pharmacokinetics” - really not looking at the entire full detailed PK. PET imaging PK really is only assessing the complex’s properties and it’s too rapid to really test real *in vivo* stability as needed once this HOPO is conjugated to targeting agents – cannot extrapolate to anything beyond just the complex itself at this stage with this data. Maybe lower renal tox but side-stepped hepatic route tox. Re-routing the clearance may not eliminate tox particularly when the route is actually a bit slower. Statements regarding skeleton and other tissue uptake, etc. cannot be made with real accuracy based on just PET imaging. A full detailed biodistribution animal study is required to define that status. Lastly, probably should have also looked to see just what was excreted and what form it was in to define and confirm that the complex remained intact – simple metabolic study by HPLC analysis would work.

Our Response: We thank the referee for this detailed and very thoughtful comment. Polyaminocarboxylate complex clearance is rapid in cases where the complex maintains *in vivo* stability, and the same is true of 343-HOPO species (see biodistribution results with Eu(III) (Sturzbecher-Hoehne *et al.* Dalton Trans. 2011, 40, 8340-8346), Am(III) (Kullgren *et al.* Toxicol. Mech. Meth. 2013, 23, 18-26), and Cm(III) (Sturzbecher-Hoehne *et al.* Chem. Euro. J. 2014, 20, 9962-9968--all of which indicate M(III)-343-HOPO complexes are stable and clear within 24 hours). We agree that we are likely not seeing the full pharmacokinetic picture in the timeframe of a small molecule complex; however, the clearance pathway and any organ uptake of a HOPO conjugate matches what we have observed here (see Deri *et al.* Bioconjugate Chem. 2015, 26, 2579-2591 for HOPO conjugate (with ⁸⁹Zr) biodistribution data). As mentioned above, obtaining biodistribution data is currently not possible due to our lab being closed until further notice as a result of Covid-19, but this is something we intend to explore in-depth in both small-molecule and bifunctional chelate systems once we have the opportunity to resume research activities.

Action Taken: We have added context and several references in the section describing the *in vivo* imaging experiments on page 10 of the manuscript: “In addition, the *in vivo* stability of metal complexes of 343-HOPO formed with both trivalent (Eu, Am, Cm) and tetravalent (Zr, Pu) metals has been investigated in multiple studies, using *ex vivo* radioanalytical techniques and *in vivo* PET imaging. All of these biodistribution studies have highlighted the unusual *in vivo* stability and quantitative, rapid hepatobiliary clearance of 343-HOPO complexes, as well as minimal or undetectable non-targeted uptake of metal ions, when compared to other commonly used chelators, including diethylenetriaminepentaacetic acid (DTPA) and desferrioxamine B (DFO).^{22,37,39,42}”

Page 12 – conclusion is overstated for what is really an incomplete preliminary study – really have not done the definitive study even at this stage to demonstrate basic stability *in vivo* of the complexes of either Sc(III) or Y(III) radionuclides. That will have to be done as well as done again once this agent is conjugated or incorporated into any targeting agent.

Our Response: We must respectfully disagree with the referee's characterization of this manuscript. While initial experiments described here may have to be re-done as part of future efforts, the results described in this work provide justification for doing further studies. We certainly agree that biodistribution studies will be critical for any new bioconjugates molecule employing the 343-HOPO ligand, however, this is much beyond the scope of this manuscript. As no suggestions for improvements to the conclusion are included, no action was taken.

Action Taken: None

REVIEWERS' COMMENTS:

Reviewer #1 (Remarks to the Author):

The authors have responded to all reviewer requests in a satisfactory manner. The manuscript is now ready for publication.

Reviewer #2 (Remarks to the Author):

I am okay with the changes made by authors and understand some of the suggested experimental works can not be performed under current situation.

Reviewer #3 (Remarks to the Author):

No further comments. Still overstates conclusions beyond the context of the results of the performed studies. While further study is justified, the results of stability studies of unconjugated complexes provides no further information than the inherent limitations of such studies.

Reviewer(s)' Comments to Author:

Referee 1

Comments:

The authors have responded to all reviewer requests in a satisfactory manner. The manuscript is now ready for publication.

Referee 2

Comments:

I am okay with the changes made by authors and understand some of the suggested experimental works cannot be performed under current situation.

Referee 3

No further comments. Still overstates conclusions beyond the context of the results of the performed studies. While further study is justified, the results of stability studies of unconjugated complexes provides no further information than the inherent limitations of such studies.

Our Response: We appreciate the feedback and have updated the manuscript conclusions in-line with referee suggestions.

Action Taken: In conclusion paragraph, sentence putting results in larger context was modified to read, "In combination, these results indicate that $^{343}\text{HOPO}$ is likely a promising chelator for both the $^{44/47}\text{Sc}$ and $^{86/90}\text{Y}$ theranostic isotope pairs." Manuscript abstract was also updated so text is consistent throughout.